# Optimizing Antibiotic Use: Addressing Resistance Through Effective Strategies and Health Policies

**DOI:** 10.3390/antibiotics13121112

**Published:** 2024-11-21

**Authors:** Maurizio Capuozzo, Andrea Zovi, Roberto Langella, Alessandro Ottaiano, Marco Cascella, Manlio Scognamiglio, Francesco Ferrara

**Affiliations:** 1Pharmaceutical Department, Asl Napoli 3 Sud, Marittima Street 3, 80056 Ercolano, Italy; m.capuozzo@aslnapoli3sud.it; 2Ministry of Health, Viale Giorgio Ribotta 5, 00144 Rome, Italy; zovi.andrea@gmail.com; 3Italian Society of Hospital Pharmacy (SIFO), SIFO Secretariat of the Lombardy Region, Via Carlo Farini 81, 20159 Milan, Italy; roberto.langella87@gmail.com; 4Istituto Nazionale Tumori di Napoli, IRCCS “G. Pascale”, Via M. Semmola, 80131 Naples, Italy; a.ottaiano@istitutotumori.na.it; 5Department of Medicine, Surgery and Dentistry, University of Salerno, 84081 Baronissi, Italy; mcascella@unisa.it; 6Pharmaceutical Department, Asl Salerno, Salvatore Giordano Street 7, 84014 Nocera Inferiore, Italy; mn.scognamiglio@aslsalerno.it; 7Pharmacy Department, Asl Napoli 3 Sud, Dell’amicizia Street 22, 80035 Nola, Italy

**Keywords:** antimicrobial resistance (AMR), inappropriate antibiotic use, public health policies, infections, surveillance and control

## Abstract

**Background:** Antimicrobial resistance (AMR) has emerged as a significant challenge to public health, posing a considerable threat to effective disease management on a global scale. The increasing incidence of infections caused by resistant bacteria has led to heightened morbidity and mortality rates, particularly among vulnerable populations. **Main text:** This review analyzes current strategies and health policies adopted in the European Union (EU) and Italy to manage AMR, presenting an in-depth examination of approaches for containment and mitigation. Factors such as excessive prescriptions, self-medication, and the misuse of antibiotics in livestock contribute to the selection and spread of resistant strains. Furthermore, this review provides a detailed overview of resistance mechanisms, including enzymatic inactivation, reduced permeability, efflux pump activity, and target site protection, with specific examples provided. The review underscores the urgent need to develop new antibiotics and implement diagnostic testing to ensure targeted prescriptions and effectively combat resistant infections. Current estimates indicate that AMR-related infections cause over 60,000 deaths annually in Europe and the United States, with projections suggesting a potential rise to 10 million deaths per year by 2050 if current trends are not reversed. The review also examines existing public health policies in Europe and Italy, focusing on national and regional strategies to combat AMR. These include promoting responsible antibiotic use, improving surveillance systems, and encouraging research and development of new therapeutic options. **Conclusions:** Finally, the review presents short- and long-term perspectives from the authors, suggesting actionable steps for policymakers and healthcare providers. Ultimately, a coordinated and multidisciplinary approach involving healthcare professionals, policymakers, and the public is essential to mitigate the impact of AMR and ensure the effectiveness of antibiotics for future generations.

## 1. Introduction

Antimicrobial resistance (AMR) refers to the ability of bacteria to evade the effects of one or more antibiotics, often leading to multidrug resistance (MDR) [1]. This enables microorganisms to survive and replicate even in the presence of antibiotic concentrations that would normally be sufficient to inhibit or kill bacteria of the same species. The indiscriminate and sometimes inappropriate use of antibiotics has historically contributed to the emergence of resistant bacterial strains, an adaptive evolutionary phenomenon driven by selective pressure from antimicrobials [2]. Consequently, the development of new classes of antibiotics has become critical. The World Health Organization (WHO) has ranked AMR among the top three global health threats of the 21st century, alongside climate change and vaccine hesitancy [3]. A key strategy in combating AMR is therapeutic appropriateness, which refers to the rational and targeted use of antibiotics. This approach not only ensures the selection of the most appropriate drug for the specific patient and infection but also optimizes treatment duration and dosage, thereby minimizing the risk of selecting resistant strains [4]. Prudent antibiotic use is essential for curbing the spread of bacterial resistance, which could otherwise further complicate the management of infections. The implementation of appropriate therapeutic protocols and stringent public health policies regarding antibiotic use are critical in the fight against AMR [5]. The global public health crisis posed by AMR is exacerbated by the rapid decline in the effectiveness of an increasing number of antibiotics and the slowdown in the development of new antimicrobial agents. This, combined with inappropriate use of existing antibiotics, has led to increased healthcare costs, with more prolonged and complex treatments, and has significantly impacted public health outcomes [6]. Despite efforts from the scientific community and pharmaceutical industries, the pipeline of new antibiotics remains insufficient to keep pace with the rapid spread of bacterial resistance. Furthermore, globalization and international trade, as well as migration and travel, facilitate the spread of resistant strains, making AMR a transnational public health challenge [7]. Tackling this issue requires coordinated global action, with close collaboration between governments, health institutions, and the scientific community. Therefore, therapeutic appropriateness becomes a fundamental pillar not only in the clinical management of infections but also in limiting the further dissemination of resistance.

## 2. Mechanisms of Antibiotic Resistance

The issue of antibiotic resistance primarily lies in acquired resistance, which occurs when bacteria that were previously susceptible to a specific antibiotic develop mechanisms to evade its action [8]. This phenomenon can arise either from chromosomal mutations, leading to so-called “chromosomal resistance”, or from the acquisition of genetic material from other bacterial populations, whether related to the recipient species or not, known as “transferable resistance” [9,10]. Chromosomal mutations, while vertically transmitted from one bacterial generation to the next, are rare events, as they are often corrected by cellular DNA repair mechanisms. However, transferable resistance poses a far more significant threat. It involves the mobilization of resistance genes between bacterial populations through mobile genetic elements such as plasmids, integrons, and transposons [11]. The transfer of these genes can occur between chromosomal and extra-chromosomal DNA, as well as between bacteria of the same or different species, via horizontal gene transfer mechanisms like transformation, conjugation, and transduction [12,13]. These processes enable recipient bacteria to rapidly acquire the ability to implement one or more sophisticated resistance mechanisms, thereby neutralizing the antibiotic’s efficacy. Such mechanisms can be classified based on the biochemical pathways involved [14]. Among the most relevant are the ability to chemically modify the antibiotic molecule, prevent its access to target sites, modify, protect, or even replace these key sites, or ultimately bypass them altogether (Figure 1).

The rapid and complex adaptation of bacteria underscores the critical importance of thoroughly understanding the dynamics of how these resistance strategies spread in order to develop effective countermeasures in the fight against antibiotic resistance—one of the most urgent challenges in modern biomedicine. Antibiotic resistance in bacteria manifests through a variety of complex and fascinating mechanisms that enable them to evade the effects of antibiotics. One of the most common mechanisms is the enzymatic inactivation of the antibiotic molecule, which can occur through both chemical modifications and the direct destruction of the molecule itself (a) [15]. For instance, bacteria produce enzymes such as beta-lactamases that hydrolyze the beta-lactam ring of penicillin and related antibiotics, rendering them inactive. Extended-spectrum beta-lactamases (ESBLs) and carbapenemases have become particularly concerning due to their broad spectrum of activity, including resistance to penicillin, cephalosporins, and even carbapenems. Gram-negative and Gram-positive bacteria, for instance, produce enzymes that chemically alter the antibiotic, such as acetylation, phosphorylation, or adenylation, thereby preventing its binding to ribosomes and compromising protein synthesis inhibition [16]. In addition, bacteria can reduce antibiotic access to its target by decreasing membrane permeability (b) or actively expelling the molecule through efflux pumps (c). Gram-negative bacteria are particularly adept at regulating membrane porins to control permeability, which allows them to restrict entry of antibiotics like beta-lactams, tetracyclines, and fluoroquinolones. Alterations in porin channels, such as OmpF (Outer Membrane Protei F) and OmpC in *E. coli*, or downregulation of porins like OprD in *Pseudomonas aeruginosa*, are classic strategies to minimize antibiotic entry [17]. Efflux pumps, present in both Gram-positive and Gram-negative bacteria, serve as another defense mechanism, expelling specific antibiotics, such as Tet pumps for tetracycline, or a broad range of drugs, as seen in MDR bacteria. Efflux pumps can be specific or MDR systems; the latter can expel a wide array of antibiotics and contribute significantly to cross-resistance. Examples include the AcrAB-TolC efflux pump in *E. coli*, which can expel beta-lactams, fluoroquinolones, and other antibiotic classes. Another bacterial strategy is to protect the target site, preventing the antibiotic from binding [18]. This is achieved through the production of specific proteins that compete for the binding site, or that remove the antibiotic from its target, as observed in resistance to tetracyclines and fluoroquinolones [19]. For instance, in tetracycline resistance, bacteria produce TetM or TetO proteins that interact with the ribosomal binding sites, effectively displacing tetracycline and allowing protein synthesis to continue. Similarly, in fluoroquinolone resistance, specific target protection proteins can bind to DNA gyrase or topoisomerase, preventing the antibiotic from inhibiting DNA replication. Additionally, bacteria can modify the target (d) through point mutations in genes or enzymatic action, as seen in resistance to macrolides and lincosamides [20]. Finally, bacteria can develop resistance by replacing the antibiotic target with new biochemical structures that are insensitive to the drug, as occurs with resistance to methicillin and vancomycin [21]. Another tactic involves the overproduction of the target, allowing the bacterium to survive even in the presence of the antibiotic, as seen in resistance to trimethoprim-sulfamethoxazole [22]. These well-orchestrated and diverse mechanisms render bacteria formidable adversaries in the fight against infections, necessitating innovative solutions to combat the growing threat of antibiotic resistance.

## 3. Causes of Antibiotic Resistance

The issue of antibiotic resistance is inherently complex, driven by a multitude of factors that interact at both individual and societal levels (Figure 2).

One of the most significant contributors to this global challenge is the inappropriate use of antibiotics, which has far-reaching consequences not only for individual patients but also for public health [23]. The misuse of antibiotics exacerbates the selective pressure on bacterial populations, leading to the emergence of resistant strains that compromise treatment efficacy and increase the risk of superinfections [24]. These resistant microorganisms often spread within healthcare settings and throughout the community, amplifying the problem. In clinical practice, one of the main drivers of excessive antibiotic use is the reliance on empirical treatment by physicians. In cases where bacterial infections present immediate life-threatening risks, there is often no time to await a precise diagnosis regarding the infectious agent and its antimicrobial susceptibility [25]. Standard diagnostic procedures, which involve extensive laboratory testing, can take days or even weeks to yield results [26]. Consequently, clinicians frequently resort to broad-spectrum antibiotic therapies or combine multiple antibiotics in the hope that one will be effective against the unidentified pathogen [27]. While this strategy can be justified in urgent cases, particularly in controlled hospital environments, it underscores the critical need for the development of rapid and accurate diagnostic tools that could facilitate more targeted and judicious antibiotic use [28]. Beyond hospital settings, the overprescription of antibiotics by general practitioners represents a more pervasive concern [29]. Many physicians, when faced with infectious symptoms in patients, opt to prescribe antibiotics based on clinical experience and local epidemiological data, often foregoing laboratory confirmation of the pathogen’s identity and its resistance profile. This practice, while seemingly pragmatic in routine outpatient care, increases the likelihood of treatment failure and necessitates the prescription of additional antibiotics, thereby intensifying selective pressure on the patient’s microbiota and promoting the spread of resistance [30]. Adding to this complexity is the behavior of patients, who frequently engage in self-directed antibiotic use. Despite regulatory frameworks requiring medical prescriptions, a substantial proportion of the population continues to access and consume antibiotics without consulting healthcare professionals [31]. For instance, an intriguing study reported that in 2019, the sale of antibiotics without a prescription in community pharmacies accounted for over 60% of global consumption [32]. This phenomenon is driven both by patient demands and, in some cases, by recommendations from pharmacy personnel. Self-medication, whether through leftover antibiotics from previous treatments or drugs acquired from family or friends, is common and often results in inappropriate dosing, suboptimal treatment duration, or the use of antibiotics that are ineffective for the specific infection [33]. The growing availability of antibiotics via online platforms presents a further challenge to efforts aimed at controlling antibiotic use. The ability to purchase antibiotics without a prescription or following superficial online consultations has led to an increase in self-medication and has further complicated the regulation of antibiotic consumption. Online antibiotics are frequently provided without sufficient oversight, and the lack of professional diagnosis undermines the quality of care received [34]. It is important to note that the extent of these behaviors varies considerably across countries, with developing nations often facing more severe challenges due to weaker regulatory oversight, inadequate prescribing practices, and limited access to qualified pharmacists. This variability highlights the need for tailored interventions that address both the systemic and behavioral drivers of antibiotic misuse in different regions.

## 4. Epidemiological and Economic Impact

AMR is emerging as one of the most serious challenges in public health, leading to a significant increase in morbidity and mortality rates associated with infections caused by resistant bacteria [35]. Evidence indicates that infections caused by resistant microorganisms have higher mortality rates compared to those caused by susceptible pathogens [36,37]. The proliferation of resistant bacteria not only prolongs the contagious period for infected individuals but also complicates the identification of suitable alternative therapies. The growing ineffectiveness of antibiotics suggests a concerning future scenario: a return to a pre-antibiotic era where common medical procedures, such as organ transplants, chemotherapy, cesarean deliveries, and even relatively simple surgical interventions like joint replacements, could become unmanageable and hazardous. Current estimates indicate that infections caused by antibiotic-resistant bacteria result in over 60,000 deaths annually in Europe and the United States. Without targeted intervention, projections suggest that by 2050, fatalities due to AMR-related infections could escalate to approximately 10 million annually [38]. An analysis conducted by the European Centre for Disease Prevention and Control (ECDC) quantified the burden of five types of infections caused by resistant bacteria within the European Union and the European Economic Area in 2015. These assessments are based on data collected by the European Antimicrobial Resistance Surveillance Network (EARS-Net) in 2018 and measured the health impact considering the number of infections, attributable deaths, and disability-adjusted life years (DALY) associated with resistant infections. For the year 2015, approximately 670,000 cases of infections caused by resistant bacteria were estimated, resulting in about 33,000 attributable deaths and a total of 874,541 DALY [39]. These figures correspond to an incidence of 130 infections and a mortality rate of 6.44 deaths per 100,000 population, resulting in a burden of 170 DALY. This value is nearly comparable to the combined burden of highly prevalent infectious diseases such as influenza, tuberculosis, and HIV, which together reach 180 DALY per 100,000 population. This comparison underscores the critical significance and severity of the AMR phenomenon. In 2017, the World Bank conducted a study based on external research to simulate the potential future economic impact of AMR, although precise cost estimates remain uncertain. The analysis revealed that AMR will lead to both direct costs—associated with treating infections caused by resistant bacteria, including hospitalizations and second-line drugs—and indirect costs resulting from chronic illness, disability, and premature death, which will reduce productivity. Two scenarios were proposed: an optimistic “low-AMR” scenario with moderate economic impacts and a more concerning “high-AMR” scenario with severe economic consequences. In the low-AMR scenario, the World Bank projected that global GDP could shrink by 0.2% by 2030 and 1% by 2050 compared to baseline growth that excludes AMR effects. In the high-AMR scenario, the decline could be steeper, with GDP falling by 3% in 2030 and 4% in 2050. The potential economic damage from AMR over the next few decades may rival or even exceed that of the 2008 financial crisis. In terms of annual losses, the low-AMR scenario estimates a global loss of around USD 900 billion per year by 2030, rising to USD 1.8 trillion by 2050. Meanwhile, the high-AMR scenario suggests annual losses could soar to USD 2.8 trillion by 2030 and reach USD 5.5 trillion by 2050—almost triple the amount projected under the low-AMR scenario. Unlike the 2008 financial crisis, which saw a relatively quick recovery, the economic impact of AMR could take far longer to address [40,41]. The development of new drugs and vaccines might take several decades, with slower deployment in low- and middle-income countries compared to wealthier nations, further deepening global economic disparities. Delays in accessing medical solutions would likely worsen existing inequalities worldwide. In terms of healthcare expenditures, both public and private sectors are expected to face significantly higher costs. Under the high-AMR scenario, by 2050, global healthcare spending could increase by about 7%, amounting to an additional USD 1.1 trillion per year. Low-income countries would bear a heavier burden, with healthcare costs potentially rising by 24%, while middle-income countries could see increases of 14%, and high-income nations around 5%. In the low-AMR scenario, healthcare spending could exceed baseline projections by approximately USD 210 billion per year by 2030 and USD 320 billion per year by 2050 [42]. These figures far surpass the investment needed to contain AMR globally. Efficiently allocating public resources will be critical to mitigating this threat and ensuring essential healthcare services for humanity, particularly for future generations.

## 5. European and Italian Healthcare Policies: Frameworks and Implementation

Antibiotic resistance is one of the most pressing threats to global health today, and it demands immediate, coordinated action across all sectors of society. The “One Health” model encapsulates this necessary approach, integrating diverse disciplines to design and implement health programs, policies, and research aimed at combating this escalating crisis [43,44]. To tackle antibiotic resistance effectively, countries must focus on strengthening healthcare systems, ensuring access to appropriate antibiotics, promoting responsible use, enforcing strict regulations on drug production and sale, and encouraging innovative solutions [45]. Surveillance of antibiotic resistance is crucial for assessing the scale of the problem and the effectiveness of interventions. The World Health Organization’s GLASS system plays a pivotal role in global monitoring, providing data that guide national and international actions [46]. Key areas of focus include the use of antibiotics in humans and animals, resistance rates, and molecular biology studies to uncover the genetic foundations of resistance, which is critical for staying ahead of evolving threats. Public education is equally essential in this battle. Misconceptions about antibiotics are widespread, often leading to misuse and dangerous practices like self-medication. It is vital to promote responsible use by educating the public on the necessity of medical consultation before using antibiotics and highlighting alternatives for symptoms where antibiotics are ineffective [47]. Healthcare professionals, too, must be continuously educated and equipped with reliable, up-to-date information on antibiotics to improve prescribing practices. Programs such as Antimicrobial Stewardship (ASP) provide the guidance and support needed for medical professionals to optimize antibiotic use and reduce resistance [48,49,50]. Figure 3 illustrates the key components involved in managing antibiotic resistance, which include healthcare policies, monitoring, education, and commitment from all stakeholders.

Additionally, infection prevention and control (IPC) measures are crucial in healthcare settings to prevent the spread of resistant pathogens. Effective IPC practices—such as proper hygiene, isolation of infected patients, and thorough sterilization of medical equipment—can significantly reduce transmission rates and protect vulnerable populations [51]. Perhaps most urgently, we must invest in research and development to discover new antibiotics and vaccines. Despite the critical need, progress is often stifled by scientific challenges and insufficient funding. Initiatives like the Transatlantic Taskforce on Antimicrobial Resistance (TATFAR) provide strategic incentives, but much more must be performed to accelerate the development of novel antimicrobial treatments [52]. Without these innovations, the world risks returning to a time when even minor infections could prove fatal. The European Union has adopted a coordinated approach to address the growing threat of AMR, with various bodies working together to monitor and combat this critical issue [53,54]. Among these, the ECDC, established in 2004, plays a key role [55]. Through advanced surveillance tools, the ECDC helps preserve the effectiveness of antibiotics by promoting prudent use and providing annual updates on the spread of AMR. Recent data suggest that AMR is responsible for over 33,000 deaths annually in Europe alone, with substantial social and economic costs—many of which could be mitigated with targeted interventions. The ECDC conducts systematic surveillance, enabling early identification of public health risks and the development of control systems within member states [56]. The resources provided by the ECDC, including strategies, guidelines, and training courses, are essential for supporting healthcare professionals in preventing and controlling AMR and healthcare-associated infections (HAIs) [57]. Two critical networks managed by the ECDC are the EARS-Net and the European Surveillance of Antimicrobial Consumption Network (ESAC-Net). EARS-Net collects comparable, accurate data on antimicrobial resistance across Europe, enabling the analysis of geographical and temporal trends [58]. These data highlight the declining efficacy of certain antibiotic classes, such as carbapenems, against resistant pathogens, particularly strains of Klebsiella pneumoniae and Acinetobacter baumannii. Infections caused by resistant bacteria pose an increasing threat, limiting available treatment options and raising morbidity and mortality rates in hospitalized patients. Surveillance data are vital in raising awareness within the scientific and political communities and supporting evidence-based health policy discussions. Simultaneously, ESAC-Net monitors antimicrobial consumption, providing member states with crucial data to assess their progress toward more rational use of these drugs. Recent analyses show a decrease in total antibiotic consumption in some countries, though significant disparities persist across Europe, with some regions still exhibiting excessive use in both hospital and community settings [59]. The European Committee on antimicrobial susceptibility testing (EUCAST) standardizes antimicrobial susceptibility testing across member states, contributing to the harmonization of AMR monitoring methodologies [60]. The guidelines published by EUCAST in 2017, for example, serve as a reference for identifying clinically and epidemiologically relevant resistance mechanisms, making it easier to monitor and understand AMR. This harmonized approach is essential, as the WHO estimates that coordinated global action could prevent 1.5 million deaths annually by 2050 [61,62]. The European Food Safety Authority (EFSA), together with the European Medicines Agency (EMA) and the ECDC, provides scientific support in tackling AMR within the food chain and animals. Overuse of antibiotics in animal husbandry is considered one of the major drivers of AMR spread, and data collected by EFSA on zoonotic diseases have highlighted an increase in resistance to pathogens transmitted from animals to humans. EFSA’s expert panels, based on annual data analysis, recommend a gradual reduction in antibiotic use in livestock, aiming to limit the emergence of resistance that could compromise food safety [63]. Lastly, the ECDC has coordinated the European Antibiotic Awareness Day (EAAD) since 2008, held annually on November 18, which aims to increase awareness among both the public and healthcare professionals about the appropriate use of antibiotics. Despite growing efforts, data indicate that 30% of antibiotics prescribed in Europe remain inappropriate, contributing to the AMR problem [64]. Initiatives like EAAD are essential in promoting responsible behaviors and reinforcing the commitment to the rational use of antibiotics, which remains a key tool in effectively combating the spread of AMR. Following global healthcare policies set by the WHO and European guidelines, Italy has undertaken significant efforts to combat AMR. Spearheaded by the Ministry of Health and national institutions; these initiatives focus on both prevention and control. In 2017, Italy introduced its National Action Plan on Antimicrobial Resistance (PNCAR) 2017–2020, following the multisectoral “One Health” approach, which integrates human, animal, and environmental health [65,66]. The plan, recently updated for 2022–2025, aims to provide a coordinated and sustainable national strategy to tackle AMR. At the core of Italy’s AMR response is its participation in European surveillance networks, particularly EARS-Net, coordinated by the European Centre for Disease Prevention and Control. This integration allows Italy to contribute data to a broader European framework, improving its capacity for timely and accurate AMR monitoring. Italian hospitals have reported concerning trends, such as the rise in resistance to carbapenems and colistin, particularly in Klebsiella pneumoniae, a major threat identified across Europe. These trends emphasize that the AMR challenge in Italy mirrors the broader European landscape, highlighting the urgency of a united approach. Italy’s Ministry of Health, through the National Centre for Disease Prevention and Control (CCM), funds a variety of AMR-related projects, fostering collaboration among central bodies, regions, and research institutions [67]. These efforts align with European goals, aiming to harmonize data collection and produce high-quality, comparable results. One of the key accomplishments has been the establishment of a national database that aggregates AMR data from regional surveillance networks, which is subsequently uploaded to TESSy, the European surveillance system [68,69]. This strengthens Italy’s role in EARS-Net and supports European efforts to generate an accurate AMR profile. The AR-ISS network, coordinated by the Istituto Superiore di Sanità (ISS), plays a crucial role in collecting data from microbiology laboratories across Italy [70,71]. These laboratories voluntarily report antibiotic resistance trends from both healthcare and community settings, targeting key pathogens of epidemiological significance. These data are integrated into EARS-Net, positioning Italy as an important contributor to the European AMR strategy. The Italian Medicines Agency (AIFA) complements this work by monitoring antibiotic consumption through its OsMed Reports, which collect data from nearly half of the country’s population [72,73,74]. These reports reveal considerable regional differences in antibiotic use, with southern Italy displaying higher usage rates compared to the north. This geographical disparity is mirrored in AMR prevalence, highlighting the need for targeted interventions. Despite progress, Italy faces persistent challenges in reducing unnecessary antibiotic prescriptions—estimates suggest that approximately 30% of antibiotics prescribed remain inappropriate. The Ministry of Health, AIFA, and ISS have responded by developing national guidelines aimed at promoting best practices in infection control and antibiotic stewardship [75]. Public awareness and healthcare provider education are crucial elements of this strategy. Italy actively participates in international campaigns like the EAAD and WHO’s World Antibiotic Awareness Week, aiming to reduce unnecessary antibiotic consumption [76]. In the agricultural and veterinary sectors, Italy has made notable strides in reducing antibiotic use, aligning with European Union efforts to prevent AMR transmission through the food chain. Data from the EFSA highlight Italy’s improvements in this area, although further reductions are necessary to meet the ambitious targets set in PNCAR 2022–2025. Italy’s ongoing efforts to combat AMR are also supported by its participation in the WHO’s “Clean Care is Safer Care” initiative, which promotes hand hygiene and infection prevention to reduce HAIs [77,78]. Participating hospitals have implemented rigorous monitoring systems to ensure compliance with hand hygiene protocols, which is crucial for controlling the spread of resistant pathogens such as Methicillin-Resistant *Staphylococcus aureus* (MRSA) [79,80]. Educational materials and monitoring programs enhance adherence to infection control measures among healthcare workers. Overall, Italy’s AMR policies reflect a comprehensive, multi-sectoral approach that is closely aligned with European strategies. Through continuous surveillance, data integration with European databases, and a focus on public and professional education, Italy remains at the forefront of AMR control efforts. Table 1 summarizes the key studies already discussed in the text, compiling essential data on antibiotic resistance and providing a comparative overview of the available evidence. This summary highlights critical trends and some regional differences, offering a foundational perspective on the progression of resistance and the pressing need for targeted interventions. In the short term, Italy’s focus will likely remain on enhancing surveillance and compliance with infection control guidelines, as well as refining stewardship programs to improve the appropriateness of antibiotic prescriptions, especially in regions with higher usage rates. Continued efforts to monitor antibiotic consumption and promote rational use will be essential to mitigate the immediate risk posed by AMR. Programs aimed at increasing awareness among healthcare providers and the public are critical to reinforcing responsible behaviors in antibiotic usage. Italy’s active participation in European surveillance networks, combined with local initiatives like the AR-ISS network, will help build a more responsive national framework that can detect emerging resistance trends in real time. In the long term, Italy aims to build a more sustainable, multi-sectoral strategy to address AMR, aligning with the “One Health” approach that considers human, animal, and environmental health collectively. A key long-term goal will be reducing antibiotic use in animal husbandry and food production further, aligning with EU recommendations to limit AMR spread through the food chain. Strengthening research and innovation within this field will also be crucial. For instance, developing alternative treatment options, such as bacteriophage therapy, and improving diagnostic tools to rapidly identify resistance patterns could transform AMR management. Continued investments in education, policy enforcement, and international collaboration will be essential to maintain progress against AMR, securing both public health and food safety for the future.

## 6. Conclusions

Antimicrobial resistance represents a global emergency of alarming proportions, necessitating immediate and collective attention. We can no longer ignore or underestimate the severity of this crisis; it is imperative to engage all sectors and maximize every available resource. Public awareness of the critical importance of antibiotics and their vulnerability due to the emergence of resistant pathogens is essential to tackling this challenge. It is time to transform the perception that AMR is an isolated problem into a profound understanding of its global nature: it transcends borders and affects every individual, with the potential to grow exponentially if not addressed with determination. Investing today in research, training, and the promotion of responsible practices is the only sustainable strategy to ensure a future where antibiotics can continue to save lives. Particular attention must be given to low- and middle-income countries, which cannot be overlooked in this collective effort. International cooperation is essential; without a united front, the fight against AMR risks becoming increasingly complex and ultimately unmanageable. Only through coordinated, multisectoral global commitment can we hope to mitigate this emergency and safeguard public health for future generations.

## Figures and Tables

**Figure 1 antibiotics-13-01112-f001:**
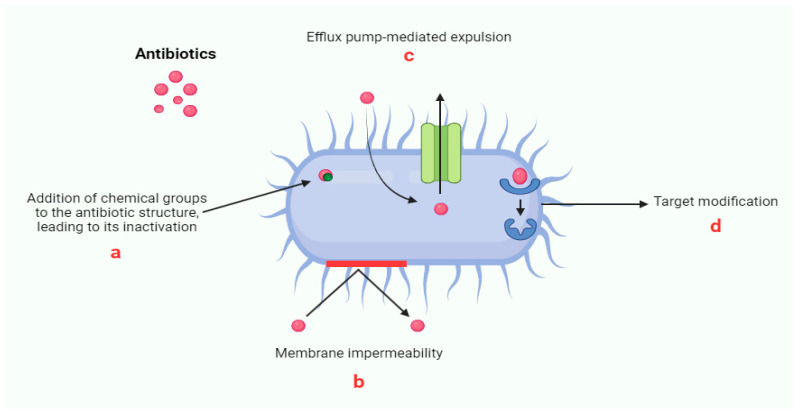
Schematic representation of the main mechanisms of antibiotic resistance.

**Figure 2 antibiotics-13-01112-f002:**
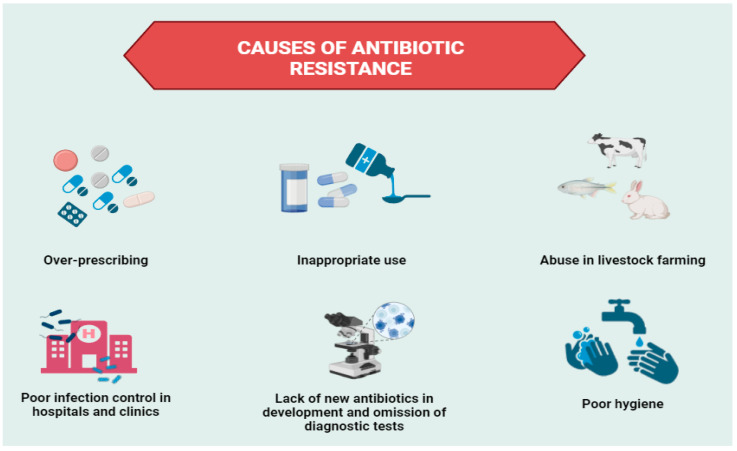
Schematic representation of major factors contributing to antibiotic resistance.

**Figure 3 antibiotics-13-01112-f003:**
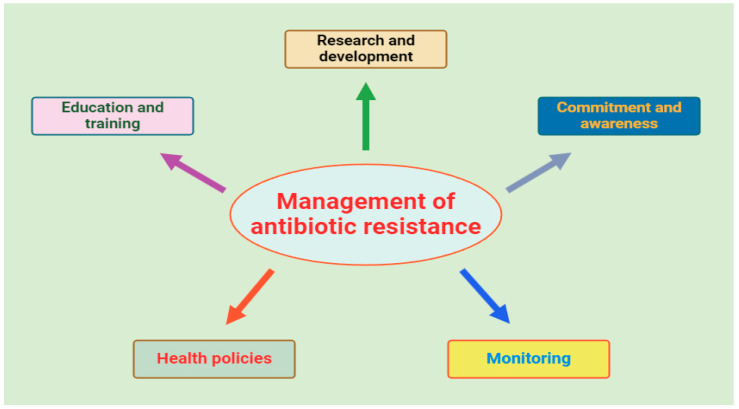
Principal guidelines for antibiotic resistance control strategies.

**Table 1 antibiotics-13-01112-t001:** Comparative overview of key data from studies on antibiotic resistance trends discussed in the text.

Study Author(s), Ref	Key Results	Implications
Alessandro Cassini at al. [39]	The estimated burden of antibiotic-resistant infections in the EU and EEA for 2015 was 671,689 infections, with 63.5% associated with healthcare. These infections led to approximately 33,110 attributable deaths and 874,541 DALYs. The burden was highest in infants (<1 year) and individuals aged 65 and older, with the greatest impact observed in Italy and Greece.	The increased burden of antibiotic-resistant infections highlights the need for prioritizing public health interventions, particularly for vulnerable populations such as infants and the elderly. Regional differences, like the higher burden in Italy and Greece, highlight the need for country-specific interventions.
World Bank Projection, 2017 [42]	Two scenarios were proposed: a “low-AMR” scenario with moderate economic impacts, projecting a 0.2% global GDP decline by 2030 and $900 billion annual losses by 2030. In the “high-AMR” scenario, global GDP could fall by 3% by 2030, with losses reaching $2.8 trillion annually. Healthcare spending could rise by 7% globally by 2050, with low-income countries facing the highest increases.	AMR could cause economic damage similar to the 2008 financial crisis, with long-term global effects. Delays in treatment development, especially in low-income countries, will worsen global inequalities. Efficient resource allocation and public health interventions are crucial to mitigate AMR and ensure equitable healthcare.
Andrea Maugeri et al. [58]	AMR was lower in countries with better indexes (*p* < 0.001), though not always linked to lower antibiotic consumption. Increased governance significantly reduced both antibiotic use (*p* < 0.001) and AMR (*p* = 0.006), with governance influencing AMR mainly through antibiotic consumption (31.5% effect).	The findings suggest that poor governance may contribute to high AMR levels, and reducing antibiotic use alone is insufficient to address AMR. Interventions to improve governance efficiency are essential for combating AMR at a global level.
Khin Hnin Pwint et al. [59]	Antibiotic consumption in public hospitals decreased by 19% from 2014 to 2017. First-line antibiotics increased (42% to 54%), while broad-spectrum antibiotics decreased (46% to 38%). Quinolone use decreased, and only linezolid, a last-resort antibiotic, was procured.	Antibiotic consumption decreased in Myanmar’s public hospitals, providing a baseline for developing an antibiotic consumption surveillance system.
Robin Vanstokstraeten et al. [61]	Between 2005 and 2017, E. coli blood isolates’ susceptibility to amoxicillin/clavulanic acid decreased from 90% to 50%. In 237 isolates, EUCAST and CLSI methods disagreed in 45% of cases, with EUCAST identifying more resistant strains in 94% of the discrepant results. EUCAST testing correlated better with the presence of beta-lactamase genes.	The study highlights low agreement between EUCAST and CLSI methods, with EUCAST identifying more resistant strains. This underscores the need for standardized testing and alignment with resistance mechanisms, like beta-lactamase genes.
Antonia Sánchez-Bautista et al. [62]	Applying EUCAST breakpoints, aminoglycoside susceptibility in Gram-negative bacilli, especially Pseudomonas aeruginosa (23.2%), decreased, while aztreonam susceptibility also declined. Resistance to clindamycin (51.5%) and gentamicin (43.1%) increased in Staphylococcus aureus.	Switching from CLSI to EUCAST criteria changes resistance percentages and alters local resistance epidemiology. A multidisciplinary approach is needed to assess the impact on empirical treatment protocols.

## Data Availability

Not applicable.

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
