# Peer review of "Optimizing Antibiotic Use: Addressing Resistance Through Effective Strategies and Health Policies"

_antibiotics, 2024, doi:10.3390/antibiotics13121112_

Round 1
Reviewer 1 Report
Comments and Suggestions for Authors
The review is very extensive and thorough. However, it may be a bit too lengthy, as it focuses on details about well-known topics, possibly going on longer than necessary. Furthermore, this is a journal for experts, so much of the information in the initial section is already well-known to the readers.
The most interesting information is found in paragraphs 4 and 5, which could be streamlined and made more accessible, perhaps with a table of sources and contacts.
Author Response
The review is very extensive and thorough. However, it may be a bit too lengthy, as it focuses on details about well-known topics, possibly going on longer than necessary. Furthermore, this is a journal for experts, so much of the information in the initial section is already well-known to the readers.
The most interesting information is found in paragraphs 4 and 5, which could be streamlined and made more accessible, perhaps with a table of sources and contacts.
Accepted and amended. The text has been revised as much as possible considering the comments of other reviewers. The suggestion of a table created to include all references in paragraphs 4 and 5 is accepted. An attempt has been made to streamline the text even though the comments of other reviewers have led to additional details being included. Thank you very much for the other positive comments on our manuscript.

Reviewer 2 Report
Comments and Suggestions for Authors
Dear authors,
My opinions about the article titled "Optimizing Antibiotic Use: Addressing Resistance Through Effective Strategies and Health Policies" are given below:
1. The presented review article is presented under six headings except Abstract and References (1. Introduction, 2. Mechanisms of antibiotic resistance, 3. Causes of Antibiotic Resistance, 4. Epidemiological and Economic Impact, 5. European and Italian Healthcare Policies: Frameworks and Implementation, and 6. Conclusion). I think this structure is appropriate for a review study.
2. The unstructured abstract representing the article states that “it highlights the multifactorial nature of antimicrobial resistance, highlighting the role of inappropriate antibiotic use in both human medicine and agriculture.” This statement suggests that this information, which will contribute to the originality of the article, will be detailed in the text.
3. In the Introduction section, which uses seven references, antimicrobial resistance is defined; the negativity of the haphazard and inappropriate use of antibiotics is mentioned; known routine information regarding the solution of the antimicrobial resistance problem is included. I believe that it would be appropriate to include a paragraph in this section that draws attention to the relationship between agriculture, animals, and humans in the context of antimicrobial resistance.
4. Fifteen references were used, and routine information was brought together under the title "Mechanisms of antibiotic resistance" and presented to the attention of the reader. Figure 1 (Schematic representation of the main mechanisms of antibiotic resistance) used in this section provides visualization of the four basic antibiotic resistance mechanisms.
5. Under the title “Causes of Antibiotic Resistance”, which is presented with twelve references and a figure, the causes of bacterial antibiotic resistance and the problems it creates are emphasized. The figure used in this section (Figure 2. Schematic representation of major factors contributing to antibiotic resistance) summarizes the main factors contributing to antibiotic resistance.
6. The "Epidemiological and Economic Impact" section, which uses eight references, provides sufficient information about the epidemiology of antibiotic resistance, its human health, and economic disadvantages.
7. In the section "European and Italian Health Policies: Frameworks and Implementation", which is probably due to a typo, but appears as 36 references but 38 references are used, the importance of the "One Health" model is emphasized and information is provided about antibiotic resistance policies in Europe and Italy as a European country. Figure 3, presented in this section, (Principal guidelines for antibiotic resistance control strategies) schematizes the five main elements of antibiotic resistance management. References 49 and 73 were not included in this section. This problem can be resolved by writing the references "[48,50]" at the end of lines 253 and 254 as "[48-50]" and the references "[72,74]" at the end of lines 342-344 as "[72-74]".
8. The "Conclusion" section emphasizes the global importance of antimicrobial resistance and the need for international cooperation in dealing with this problem; In the light of the information compiled, suggestions are made on how to overcome this problem. This section is sufficient to provide results for the purpose of the review.
9. In a review article with seven authors, four of the authors are related to the field of pharmacy, and in an article on the problem of antibiotic resistance, the absence of any clinical expert in the field of Infectious Diseases or Medical Microbiology weakens the scientific strength of the presented study. Again, although the summary mentions the relationship between inappropriate antibiotic use in both human medicine and agriculture and antimicrobial resistance, it was observed that there was no information in the text that draws attention to the relationship between antimicrobial resistance, agriculture, animals and humans. The original aspect of this article, which mostly repeats known results, is the information presented in the "European and Italian Healthcare Policies: Frameworks and Implementation" section on an Italian scale.
As a result, I believe that in order for this review article to be published in a Q1 group journal, the article should be reorganized by taking into account the limitations and corrections mentioned above.
Best Regards
Author Response
My opinions about the article titled "Optimizing Antibiotic Use: Addressing Resistance Through Effective Strategies and Health Policies" are given below:
- The presented review article is presented under six headings except Abstract and References (1. Introduction, 2. Mechanisms of antibiotic resistance, 3. Causes of Antibiotic Resistance, 4. Epidemiological and Economic Impact, 5. European and Italian Healthcare Policies: Frameworks and Implementation, and 6. Conclusion). I think this structure is appropriate for a review study.
Thank you very much for this positive assessment of the structure of the text.
- The unstructured abstractrepresenting the article states that “it highlights the multifactorial nature of antimicrobial resistance, highlighting the role of inappropriate antibiotic use in both human medicine and agriculture.” This statement suggests that this information, which will contribute to the originality of the article, will be detailed in the text.
Accepted and amended. This sentence is a typo. It was our intention to also address the animal and agricultural aspects, however the text became much longer so this chapter was not addressed in this review. The sentence is deleted so as not to give the reader false expectations. The abstract is made more relevant to the content and has also been structured into ‘Background’, ‘Main text’ and ‘Conclusion’.
- In the Introductionsection, which uses seven references, antimicrobial resistance is defined; the negativity of the haphazard and inappropriate use of antibiotics is mentioned; known routine information regarding the solution of the antimicrobial resistance problem is included. I believe that it would be appropriate to include a paragraph in this section that draws attention to the relationship between agriculture, animals, and humans in the context of antimicrobial resistance.
Accepted and clarified. In agreement with other reviewers, the part of the introduction was already quite long, so it was shortened further to leave more space for the government impact chapters that are crucial in the definition of this review article in counteracting resistance. For these reasons, the animal and agricultural aspects are not mentioned as there is no possibility of developing them in a text with an already over 6000 words.
- Fifteen references were used, and routine information was brought together under the title "Mechanisms of antibiotic resistance" and presented to the attention of the reader. Figure 1 (Schematic representation of the main mechanisms of antibiotic resistance) used in this section provides visualization of the four basic antibiotic resistance mechanisms.
Accepted and improved. The text in agreement also with other reviewers was redefined and improved in its scope.
- Under the title “Causes of Antibiotic Resistance”, which is presented with twelve references and a figure, the causes of bacterial antibiotic resistance and the problems it creates are emphasized. The figure used in this section (Figure 2. Schematic representation of major factors contributing to antibiotic resistance) summarizes the main factors contributing to antibiotic resistance.
This paragraph is confirmed for its concise and intuitive purpose, which aids the reader in understanding the other chapters.
- The "Epidemiological and Economic Impact" section, which uses eight references, provides sufficient information about the epidemiology of antibiotic resistance, its human health, and economic disadvantages.
This paragraph is confirmed for its concise and intuitive purpose, which aids the reader in understanding the other chapters.
- In the section "European and Italian Health Policies: Frameworks and Implementation", which is probably due to a typo, but appears as 36 references but 38 references are used, the importance of the "One Health" model is emphasized and information is provided about antibiotic resistance policies in Europe and Italy as a European country. Figure 3, presented in this section, (Principal guidelines for antibiotic resistance control strategies) schematizes the five main elements of antibiotic resistance management. References 49 and 73 were not included in this section. This problem can be resolved by writing the references "[48,50]" at the end of lines 253 and 254 as "[48-50]" and the references "[72,74]" at the end of lines 342-344 as "[72-74]".
Accepted and amended. We thank the reviewer for the valuable suggestion. We have adjusted the numbering and references. The text has also been implemented and improved with final considerations requested by another reviewer.
- The "Conclusion" section emphasizes the global importance of antimicrobial resistance and the need for international cooperation in dealing with this problem; In the light of the information compiled, suggestions are made on how to overcome this problem. This section is sufficient to provide results for the purpose of the review.
This paragraph is confirmed for its concise and intuitive purpose, which aids the reader in understanding the other chapters.
- In a review article with seven authors, four of the authors are related to the field of pharmacy, and in an article on the problem of antibiotic resistance, the absence of any clinical expert in the field of Infectious Diseases or Medical Microbiology weakens the scientific strength of the presented study. Again, although the summary mentions the relationship between inappropriate antibiotic use in both human medicine and agriculture and antimicrobial resistance, it was observed that there was no information in the text that draws attention to the relationship between antimicrobial resistance, agriculture, animals and humans. The original aspect of this article, which mostly repeats known results, is the information presented in the "European and Italian Healthcare Policies: Frameworks and Implementation" section on an Italian scale.
Accepted and amended. Regarding other comments submitted about abstract and introduction we, in agreement with other reviewers, have tried to satisfy all comments by trying to reduce the introductory parts and make the text less long about widely known parts. For these reasons, issues concerning agriculture and animals were deleted and the abstract rewritten. Chapters 4 and 5, which are crucial for the purpose of the article, fall within the competence of the pharmacist and therefore also satisfy the robustness of the study, which addresses the problem of bacterial resistance from a governmental and economic perspective as well.
As a result, I believe that in order for this review article to be published in a Q1 group journal, the article should be reorganized by taking into account the limitations and corrections mentioned above.
Accepted, modified and improved. All suggestions of the 3 reviewers were made and tried to make the text homogeneous and in accordance with the professionalism of the authors.

Reviewer 3 Report
Comments and Suggestions for Authors
The present review demonstrates the current situation and opinion in managing antibiotic resistance through strategies and health policies from EU and Italian frameworks and implementation. The manuscript was sufficiently prepared, but a more comprehensive version with extensive development in certain points should be amended. In my humble opinion, several major points as indicated below should be considerably clarified and revised.
· Abstract: The abstract should be comprehensively revised again after considering the collective comments below.
· Line 95-123: This part is very relevant to the context of this review. Thus, it would be better and recommended to provide further in-depth details about each mechanism mentioned here.
· Line 94, Figure 1: Captions, such as (a), (b), (c) or (i), (ii), (iii) …….should be labeled in the picture where it corresponds to each antibiotic-resistant mechanism described in the text.
· Line 129-173: A summary table with examples of case reports or WHO, EFSA statistics related to each type of factor affecting resistance development is highly recommended.
· Line 174-232: For a better demonstration of data, epidemiological reports should be mentioned and summarized in a Table form. Economic relevance is recommended to be depicted in certain kinds of (pie)charts.
· Line 372: A short- and long-term perspectives from the authors’ point of view should be also addressed here, before the conclusion section.

Author Response
The present review demonstrates the current situation and opinion in managing antibiotic resistance through strategies and health policies from EU and Italian frameworks and implementation. The manuscript was sufficiently prepared, but a more comprehensive version with extensive development in certain points should be amended. In my humble opinion, several major points as indicated below should be considerably clarified and revised.
Accepted and amended. The paper was revised in its entirety by focusing on all the suggestions made. A table was also created for chapters 4 and 5 to make all references with "key results" and "implications" clearer and more readable.
- Abstract: The abstract should be comprehensively revised again after considering the collective comments below.
Accepted and amended. The abstract has been updated with all the changes suggested and made. The changes are all in red in the text.
- Line 95-123: This part is very relevant to the context of this review. Thus, it would be better and recommended to provide further in-depth details about each mechanism mentioned here.
Accepted and amended. The text is completely revised and improved. All changes in red.
- Line 94, Figure 1: Captions, such as (a), (b), (c) or (i), (ii), (iii) …….should be labeled in the picture where it corresponds to each antibiotic-resistant mechanism described in the text.
Accepted, amended and added. Every suggestion has been executed. We thank the reviewer for this valuable tip.
- Line 129-173: A summary table with examples of case reports or WHO, EFSA statistics related to each type of factor affecting resistance development is highly recommended.
Accepted and clarified. We thank you for the suggested approach but nevertheless this is highly difficult and complicated to implement as the variability is very high in the factors and they change very quickly. For these reasons, the creation of a table could never frame the current situation at the time the reader is reading. It was decided to modify the text to make these aspects clear and to refer to the official WHO and EFSA websites for a current and always defined situation.
- Line 174-232: For a better demonstration of data, epidemiological reports should be mentioned and summarized in a Table form. Economic relevance is recommended to be depicted in certain kinds of (pie)charts.
Accepted and partially amended. The table was also created based on the suggestions of other authors. Regarding economic relevance we do not think it is appropriate to burden the text with information that would lengthen the text and burden the article with additional figures. In agreement also with other reviewers, we have tried to develop and detail certain aspects without digressing into other concepts that would make the text excessively long.
- Line 372: A short- and long-term perspectives from the authors’ point of view should be also addressed here, before the conclusion section.
Accepted and added. Additional text has been added in red on lines 389-407 to focus our perspective before the final conclusions.

Round 2
Reviewer 2 Report
Comments and Suggestions for Authors
Dear Authors,
I believe that it is appropriate to publish the article in which the suggested changes have been made.
Kind regards
Reviewer 3 Report
Comments and Suggestions for Authors
The manuscript has undergone significant improvements and can now be considered for publication.